# Non-Operative Management of Polytraumatized Patients: Body Imaging beyond CT

**DOI:** 10.3390/diagnostics13071347

**Published:** 2023-04-04

**Authors:** Francesca Iacobellis, Marco Di Serafino, Martina Caruso, Giuseppina Dell’Aversano Orabona, Chiara Rinaldo, Dario Grimaldi, Francesco Verde, Vittorio Sabatino, Maria Laura Schillirò, Giuliana Giacobbe, Gianluca Ponticiello, Mariano Scaglione, Luigia Romano

**Affiliations:** 1Department of General and Emergency Radiology, “Antonio Cardarelli” Hospital, Via A. Cardarelli 9, 80131 Napoli, Italy; 2Department of Clinical and Experimental Medicine, University of Sassari, 07100 Sassari, Italy; 3James Cook University Hospital, Middlesbrough TS4 3BW, UK

**Keywords:** trauma, polytrauma, non-operative management, imaging, CT, MDCT, ultrasound, CEUS, MRI

## Abstract

In the transition from the operative to the conservative approach for the polytraumatized patients who undergo blunt trauma, diagnostic imaging has assumed a pivotal role, currently offering various opportunities, particularly in the follow-up of these patients. The choice of the most suitable imaging method in this setting mainly depends on the injury complications we are looking for, the patient conditions (mobilization, cooperation, medications, allergies and age), the biological invasiveness, and the availability of each imaging method. Computed Tomography (CT) represents the “standard” imaging technique in the polytraumatized patient due to the high diagnostic performance when a correct imaging protocol is adopted, despite suffering from invasiveness due to radiation dose and intravenous contrast agent administration. Ultrasound (US) is a readily available technology, cheap, bedside performable and integrable with intravenous contrast agent (Contrast enhanced US—CEUS) to enhance the diagnostic performance, but it may suffer particularly from limited panoramicity and operator dependance. Magnetic Resonance (MR), until now, has been adopted in specific contexts, such as biliopancreatic injuries, but in recent experiences, it showed a great potential in the follow-up of polytraumatized patients; however, its availability may be limited in some context, and there are specific contraindications, such as as claustrophobia and the presence non-MR compatible devices. In this article, the role of each imaging method in the body-imaging follow-up of adult polytraumatized patients will be reviewed, enhancing the value of integrated imaging, as shown in several cases from our experience.

## 1. Introduction

The transition from the operative to the non-operative management (NOM) required new technologies and new professional skills to properly monitor the polytraumatized patients. In this setting, the diagnostic imaging assumed a pivotal role, as it became necessary to adopt non-invasive methods that are adequate to monitor the healing of the injured organs and the possible occurring complications. An ideal imaging method should ensure availability, high diagnostic accuracy, low invasiveness, low execution time, and low costs. Actually, existing imaging methods differ from each other for these properties, and each one has pros and cons, so the choice of the most suitable method depends on the injury complications to look for, the patient conditions (mobilization, cooperation, dressings, allergies, and age), the biological invasiveness, and the availability of each of them.

Computed Tomography (CT) represents the “gold standard” imaging technique in the first-line evaluation of polytraumatized patients [1,2,3], as it ensures high diagnostic accuracy, rapid execution time, and hospital availability; however, once the diagnosis is made and the patient is stable, a series of possibilities open up regarding the best diagnostic imaging tool to choose for monitoring the diagnosed lesions or detecting complications, depending on the involved anatomical structures, the grade of each injury, and the availability and expertise of each imaging method [4,5,6,7,8].

In this article, the role of each imaging method in the body-imaging follow-up of adult polytraumatized patients will be reviewed, enhancing the value of integrated imaging, as shown in several cases from our experience.

## 2. Computed Tomography (CT)

Computed Tomography (CT) constitutes the reference imaging method for polytraumatized patients mainly due to the very high diagnostic performance for whole-body injuries in a short time, when an up-to-date technology is used and a correct imaging protocol is adopted, despite suffering from invasiveness due to the radiation dose and to intravenous contrast agent administration [1,2,9,10,11].

In the setting of nonoperative management, CT is mainly adopted when the patient has multiple injuries that do now allow for an exclusive evaluation of each of them, i.e., combined brain, thoracic injuries, thoraco-abdominal injuries (Figure 1), or multiple abdominal injuries, and in case of vessel injury, initially treated conservatively, endovascularly, or surgically (Figure 2).

The imaging protocol we suggest in the CT follow-up is a multiphasic CT protocol [10,11,12], the same adopted at the admission, as new lesions may manifest in the meantime, especially vascular, and so it is important to have all the data useful to orient the treatment (Figure 2) [10,11,13,14].

It consists of a non-contrast scan of the head, followed by an arterial and a venous phase, with a single bolus injection (80–130 mL of iodinated contrast medium, according to the patient’s weight), at a high concentration (370–400 mg I/mL), injected at 3.5–5 mL/s, and followed by a 40 mL saline chaser at the same flow rate to obtain optimal vessel depiction) and two separate acquisitions. Automated bolus tracking identifies the arterial phase, a region of interest (ROI) is placed on the aortic arch, and arterial phase scanning starts when an attenuation threshold of 100 Hounsfield Unit (HU) is reached; depending on the speed of acquisition of the scanner, it may be necessary to wait a few additional seconds. The portal venous phase is performed at a 60-to-70 s delay from the beginning of the injection, and an additional late phase at 3–5 min may be required to differentiate arterial bleeding from lower pressure venous bleeding, or at 5–20 min to evaluate urinary extravasation in patients with kidney injuries [9,12].

Technological advances in the field lead to the development of post-processing techniques exploiting dual-energy technology, which offers slight advantages over traditional CT by scanning the same anatomical structures with different kilovoltages (lower energy at 80 kV or 100 kV and higher energy at 140 kV) and allowing them to improve the contrast resolution, adopting lower doses of intravenous contrast agent (about 50% less than a conventional CT) (Figure 2) [15,16]. Moreover, dual-energy CT technology allows the reconstruction of virtual non-contrast (VNC) images from a single-phase contrast-agent–enhanced acquisition, potentially reducing the need for multiphasic CT acquisition to characterize the bleeding lesion with an overall reduction in the radiation dose applied to the patient (approximately 30% less than conventional CT) [17,18].

In the setting of the non-operative management of polytraumatized patients who underwent high-energy trauma, it is reasonable and suggested to perform a routine imaging follow-up with enhanced-CT scan at about 24 h to evaluate, particularly, the evolution of possibly bleeding lesions, i.e., hematomas, hemomediastinum, hemotorax, isolated hemoperitoneum/hemoretroperitoneum, stranding mesentery or the outcome of endovascularly treated lesions, and the possible onset of findings of pancreatic or bowel trauma that may be initially ambiguous or subtle (Figure 2).

In thoracic trauma, the majority of patients can be managed conservatively [19]. In the NOM after thoracic trauma, there is not a general consensus when to follow-up the chest, usually depending on the patient clinical conditions; however, in clinical experience, they are prudentially re-evaluated in a short time (24–48 h), injuries likely to evolve, as those named before, or suspicious findings of such injuries as pneumomediastinum, extended pulmonary contusions, large pneumatoceles and hematoceles [20].

In abdominal trauma, pancreatic trauma has an evolving nature, and radiological findings may become more apparent over time, with the development of post-traumatic pancreatitis, edema, leakage of pancreatic enzymes, and subsequent auto-digestion of the parenchyma [21,22], so, after high-energy blunt trauma, it is reasonable to re-evaluate these patients in a short time.

About bowel and mesenteric traumas, several contributions in the literature agree that these lesions become symptomatic within about 9 h from the traumatic event [23]. Therefore, if there is a high suspicion regarding this kind of injury, the first follow-up CT should be planned at 8–12 h (Figure 3) [24,25].

Regarding the spleen, an imaging follow-up is particularly suggested in injuries for WSES Classes II–III, AAST Grades III–V, in the first 48–72 h, to exclude the development of vascular complications (Figure 2 and Figure 4) [6,26].

About the liver, currently there is still no consensus about the time to perform follow-up imaging after NOM, but considering that 4% of patients after traumatic liver injury may develop contained vascular injuries that are not correlated to the severity of liver injury [27], it is reasonable to re-image these patients 48–72 h after trauma (Figure 5 and Figure 6) [5,28].

It is suggested to have follow-up imaging, especially in patients at higher risks for biliary complications, such as high-grade injury, central hepatic injury and post main hepatic artery embolization, as well as in patients with the onset of non-specific abdominal complaint, developing jaundice or abruptly elevated liver enzymes [7,27].

In the case of a kidney injury, the data suggest that routine CT imaging 48 h post-trauma can be safely omitted for patients with low-grade blunt renal injury, as long as they remain clinically stable, whereas patients with high-grade renal injury have the highest risk for clinical progress; thus, close surveillance should be considered in this group (Figure 7) [29].

Thus, it is suggested to repeat the 48 h CT scan in patients with a high-grade renal injury (grades IV-V) and in patients who have signs and symptoms of complications such as high-grade fever, persistent/worsening back pain, ongoing blood loss, intermittent gross hematuria, hypertension and abdominal distension after 48 h of admission (Figure 7) [7]. Furthermore, consider that in patients with large perinephric hematoma and deep parenchymal injury, a collecting system injury may be obscured [7], and so it is prudent to re-image these patients, including in the CT protocol late-phase acquisitions (Figure 7).

Adrenal trauma in the general adult population is relatively rare, with a reported incidence varying from the range of 2–3% [30] to approximately 7% [31]. It usually results from blunt trauma and is rarely seen in penetrating trauma [31], and in a high percentage of cases, it is accompanied by other intra-abdominal, retroperitoneal, or intrathoracic injuries (Figure 8) [30,31], so these patients are usually re-imaged to re-evaluate also other injuries or may deserve a dedicated re-evaluation in the case of the angioembolization of vascular injuries. Clinical manifestations of adrenal hemorrhage are rare; however, bilateral adrenal hemorrhage may present with acute adrenal insufficiency and can be considered to be a potentially fatal condition [30]. Unilateral adrenal injuries have limited clinical significance unless they cause compression of the inferior vena cava with risk of thrombus formation or adrenal hematomas that may become overinfected [30].

The enhanced CT is the imaging method of choice to follow-up, in short time, about 24 h, those patients with suspected vessel injury or with low-grade vessel injuries treated conservatively [9,32,33,34,35,36,37].

Thus, as discussed, there are reasonable reasons and a substantial agreement in the literature to perform, in adult polytraumatized patients after blunt trauma, the first follow-up CT generally at 24–48 h from the admission CT. Whereas, for the following timing of examinations a standardized consensus does not exists, and the timing strictly depends on patient condition and kind of injuries, except for spleen and liver injuries for whom there is consensus regarding the need of re-evaluation at 7–14 days from trauma, as that is the time within almost all of the delayed vascular complications appear [5,6,27].

Other vascular injuries, such as possible complications that occur in the liver during NOM, including bilomas, liver hematomas and liver abscesses, can also be conservatively treated [38]. Splenic complications during NOM can also be represented by pseudocysts, abscesses and splenosis [38]. After renal trauma, we can observe a perinephric abscess, urine extravasation and urinoma (Figure 7), and urinary fistula [38]. In unfortunate and fortunately infrequent cases, intra-abdominal infections can negatively evolve into sepsis [39].

As enhanced CT is not feasible for all the follow-up examinations, considering the kind of lesion to follow-up, biological invasiveness and the costs, it is important to keep in mind the potentialities, advantages and limits also of the other imaging methods, such as ultrasound (US) and magnetic resonance (MR), that may play an important role in the follow-up of these patients.

## 3. Ultrasound (US)

US is a readily available technology, cheap, bedside performable, non-invasive and integrable with intravenous contrast agent (contrast-enhanced US, CEUS) to enhance the diagnostic performance, but it may suffer particularly from limited panoramicity and operator dependance, requesting highly trained personnel to avoid misinterpretation of US findings that represent a serious risk in diagnosis [40].

Due to its properties, and considering its limits, US-CEUS may be adopted in the follow-up of patients who underwent blunt abdominal trauma, particularly in cooperating patients with an adequate body habitus, without extensive cutaneous medications in the area of the examination, and in which it is requested to particularly re-evaluate a specific organ injury (Figure 4, Figure 6, Figure 8, Figure 9 and Figure 10) [8,26,40].

Indeed, CEUS is able to identify and grade traumatic parenchymal and vascular injuries in explorable organs, with sensitivity and specificity levels similar to those seen in MDCT, which reach up to 95% [8]; this is thanks to the use of intravenous contrast medium, consisting in microbubbles, that after the injection allow us to continuously appreciate its dynamic in the region of interest, during each contrast phase, in particular, the arterial and parenchymatous–venous phase. In the early arterial phase, it is possible to obtain an optimal depiction of contained vascular injuries, such as pseudoaneurysms and arteriovenous fistulas, when they occurred, whereas, in the following phases, the parenchymal enhancement can be studied, evaluating the extension of injuries and eventually the complications that occur, such as fluid collections, abscessed, bilomas and hematomas [38]. Limits exist in the evaluation of active bleeding that can actually be incidentally detected, even if this cannot be the imaging methods of choice for this purpose due to the constrained limited panoramicity and in the evaluation of urinary or biliary leaks, as the intravenous contrast medium is excreted through the lungs during breathing. Furthermore, patients with suspected active bleeding, as well as suspected bowel injury, would warrant a CT examination rather than CEUS [38,40].

The CEUS protocol consists of a previous US B-mode and Doppler evaluation to detect initial findings and to preliminarily study the anatomical site after the administration of intravenous contrast medium administration. The intravenous contrast used consists of a bolus of about 2 mL (90 μg of sulfur hexafluoride), followed by approximately 5–10 mL of saline solution administered through an antecubital vein. The flash-mode technique allows us to re-evaluate the dynamic post-contrast perfusion, emitting a short US pulse with a very high mechanical index to destroy accumulated microbubbles within an area of interest, even if with a reduced concentration of the contrast agent, prior to its excretion through the lungs. It is also possible to administer to the patient a second bolus to re-evaluate the same organ or, more commonly, to study another organ if needed, but in this case, it should be considered the persistence of the already injected contrast agent, waiting to be excreted through the lungs until 15 min [8,40,41].

During follow-up CEUS examinations, the known injured organ is targeted, and all postcontrastographic phases are evaluated to exclude any contained vascular lesions in the arterial phase. Any regression of the parenchymal injured area is monitored during the venous and late phases. In the case of any worsening changes, the use of CT with intravenous iodinated contrast medium administration is mandatory [8].

Furthermore, CEUS can be used to evaluate uncertain CT findings related to abdominal trauma during follow-up (e.g., a point-of-care CEUS) without overlap in contrast media excretion since US contrast medium is excreted by lungs [41].

A particular segment of patients requiring CEUS upon admission and in the follow-up are the male patients who underwent blunt trauma of the scrotum and penis, which represent districts that can be adequately and satisfactorily imaged by US-CEUS [42,43] and, eventually, by MRI, as second-level imaging technique [44].

Thus, CEUS may constitute a good imaging option in the follow-up of polytraumatized patient with abdominal parenchymatous organ injury and testis, too [27], considering to integrate findings with enhanced-CT or MRI when needed [4].

## 4. Magnetic Resonance (MR)

MR, until now adopted in specific contexts as biliopancreatic injuries, from recent experiences showed a great potential in the follow-up of polytraumatized patients [9], as it is as panoramic as CT, but with lower invasiveness and higher tissue contrast. MR limits are mainly related to the machine availability; the patient cooperation, considering the longer acquisition time (pain with obliged decubitus, claustrophobia); and the presence of non-compatible MR devices (orthopedic implants, cardiac implants and prosthesis).

MRI may be considered to be a different imaging option, alternative or integrative to the others in the follow-up of patients with minor and major solid organ injuries (Figure 4, Figure 9, Figure 10 and Figure 11), or as an additional tool able to solve diagnostic CT doubts, avoiding repeated radiation and contrast medium exposures, in a usually young population [4,45].

Indeed, the MR allows us to identify, characterize and monitor peritoneal-fluid collections, abscesses, hematomas, bilomas, intraparenchymal lacerations and hematomas, as well as abscesses and vascular injuries, and the signal behavior on the different sequences offers more parameters to objectify the evaluation; it has greater sensitivity in identifying traumatic lesions in patients with “isolated” hemoperitoneum at CT, thus focusing the attention and the following instrumental evaluation only on the specific site of injury and may clarify doubtful CT findings and may overcome CT artifacts present in suboptimal examination.

Different from US-CEUS, MRI may also detect and monitor biliary—with dedicated intravenous contrast media—and urinary extravasation. This has implications in the patient management and in limiting radiation exposure, unnecessary further imaging evaluations, optimizing length of stay and related costs. 

From a strictly scientific point of view, accuracy in the diagnosis of active bleeding has not yet been demonstrated on human patients, even if it is reasonable to think that it can be possible to detect active bleeding, as by now it has been demonstrated the capability to detect contained vascular injuries (Figure 11). However, patients with suspected active bleeding are correctly sent to undergo an enhanced CT to confirm the bleeding eventually present and to obtain a vascular map that is useful to guide the interventional radiologist, if needed.

Future wider opportunities to use MRI in this setting will be linked to the possibility of reducing examination times and to further optimize the sequences, with the possibility of limiting the contrast study to specific cases.

In Table 1, we summarize the advantages and recommendations and the disadvantages and contraindications of the imaging techniques that can be adopted in the follow-up of polytraumatized patients.

## 5. Conclusions

Follow-up imaging timing and modality in the NOM setting is still debated. However, in adult polytraumatized patients after they suffered blunt trauma with body injuries, particularly with abdominal parenchymal injuries non-operatively managed, the adoption of multiple integrated imaging methods may offer good opportunities in the follow-up, when clinically indicated, adding the potentialities of each one and reducing the limits and negative effects.

## Figures and Tables

**Figure 1 diagnostics-13-01347-f001:**
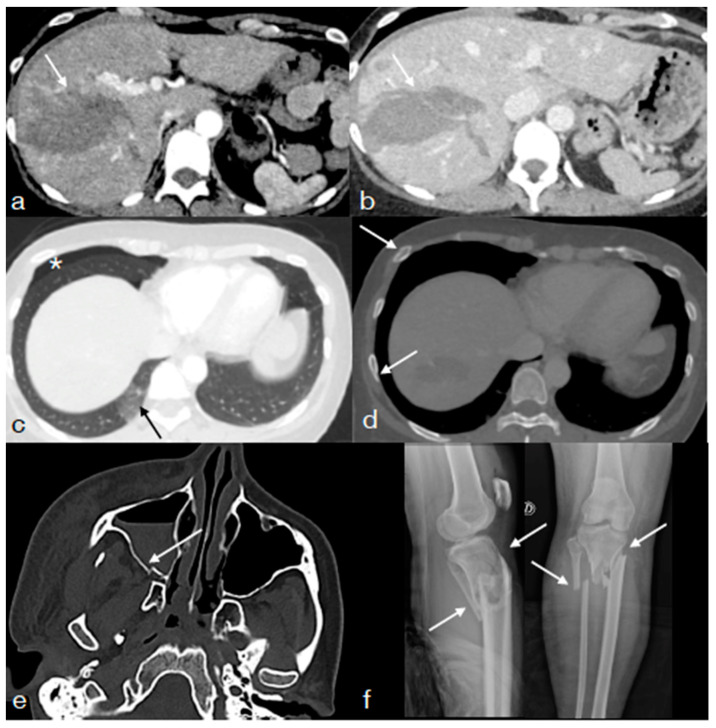
A thirty-four-year-old female underwent high-energy blunt trauma showing in the enhanced CT performed upon admission for multiple injuries: extensive liver injury, AAST IV ((**a**) arterial phase, arrow; (**b**) portal phase, arrow); pulmonary contusions ((**c**) straight arrow); low-grade pneumothorax ((**c**) asterisk); multiple rib fractures ((**d**) arrows); fracture of the maxillary sinus ((**e**) arrow); multiple leg fractures ((**f**), arrows). This patient needs to be re-imaged with CT in the first imaging follow-up to re-evaluate multiple cranio-facial and body injuries and the possible early development of vascular liver injuries.

**Figure 2 diagnostics-13-01347-f002:**
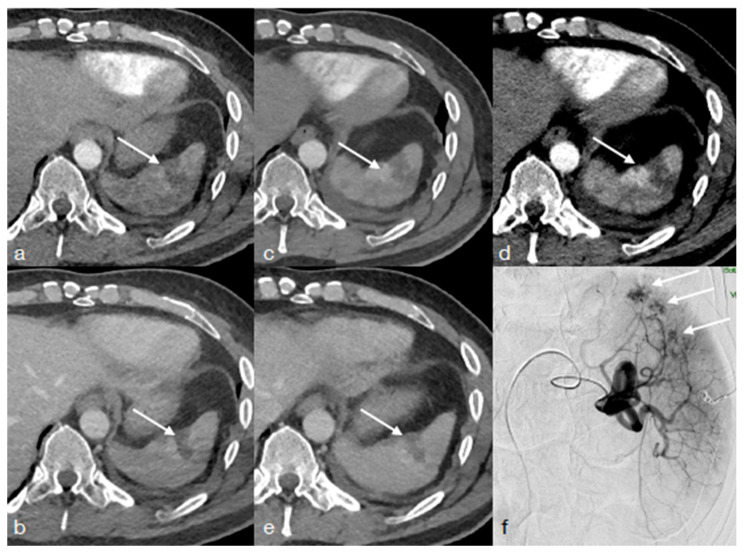
A twenty-seven year-old male underwent high-energy trauma. Enhanced CT performed upon admission in arterial (**a**) and portal phase (**b**) shows a spleen laceration ((**a**,**b**) arrows). The first follow-up CT, performed 24 h later (**c**–**e**), was acquired with dual-energy technique ((**c**) arterial phase, (**d**) low KeV arterial phase and (**e**) portal phase). Note the better visualization of the contained vascular injuries in (**d**) (arrow), in comparison with ((**c**), arrow) adjacent to the laceration (**e**, arrow). CT findings were confirmed at angiography ((**f**), arrows) and then, treated.

**Figure 3 diagnostics-13-01347-f003:**
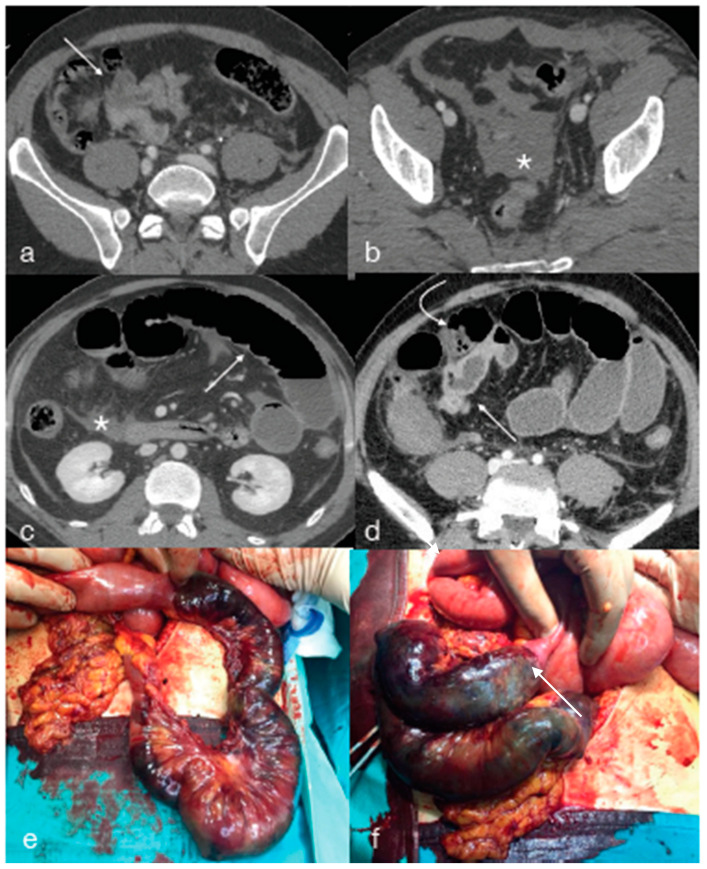
A twenty-eight-year-old male patient underwent high-energy blunt trauma. The enhanced CT performed upon admission, portal phase axial view, showed mesenteric fat stranding ((**a**) arrow) and free fluid in the pelvis ((**b**) asterisk). A few hours later, the patient developed intense abdominal pain and abdominal tension, and the enhanced CT examination was repeated ((**c**,**d**) portal phase, axial plane), showing free peritoneal fluid ((**c**) asterisk), bowel distension with air–fluid level ((**c**), arrow) proximal to an ileal tract with inhomogeneous enhancement of the wall, partly with “paper thin” appearance due to arterial ischemia ((**d**) curved arrow) and partly thickened and hyperdense ((**d**) straight arrow) due to ischemic–reperfusion changes. These findings were indicative of a mesenteric-bowel injury, so the patient underwent emergent surgery, confirming a mesenteric laceration, (**e**) causing intestinal ischemic–necrotic changes (**f**, arrows).

**Figure 4 diagnostics-13-01347-f004:**
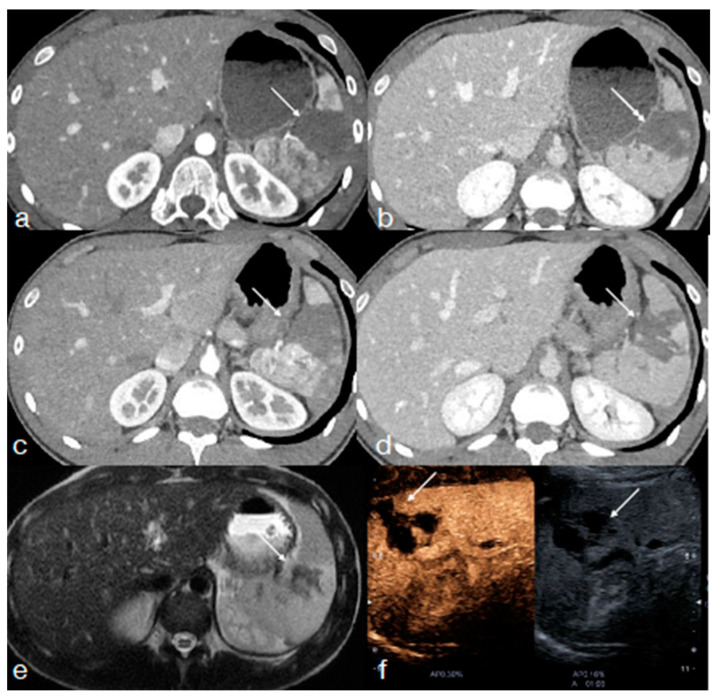
Twenty-year-old male patient underwent high-energy left-flank trauma with high-grade spleen injury, AAST IV: (**a**,**b**) enhanced CT upon admission for arterial (**a**) and portal (**b**) phases showing the spleen laceration ((**a**,**b**) arrows). Due to the high-grade splenic injury, the patient was re-evaluated at 24 h by enhanced CT ((**c**) arterial phase and (**d**) portal phase, arrows) and subsequently at 1 week by MRI ((**e**) T2W Fat Sat sequence in axial plain showing the splenic laceration, arrow) and five more days later by CEUS ((**f**) arrows), demonstrating the progressive healing of the laceration without evidence of vascular injuries.

**Figure 5 diagnostics-13-01347-f005:**
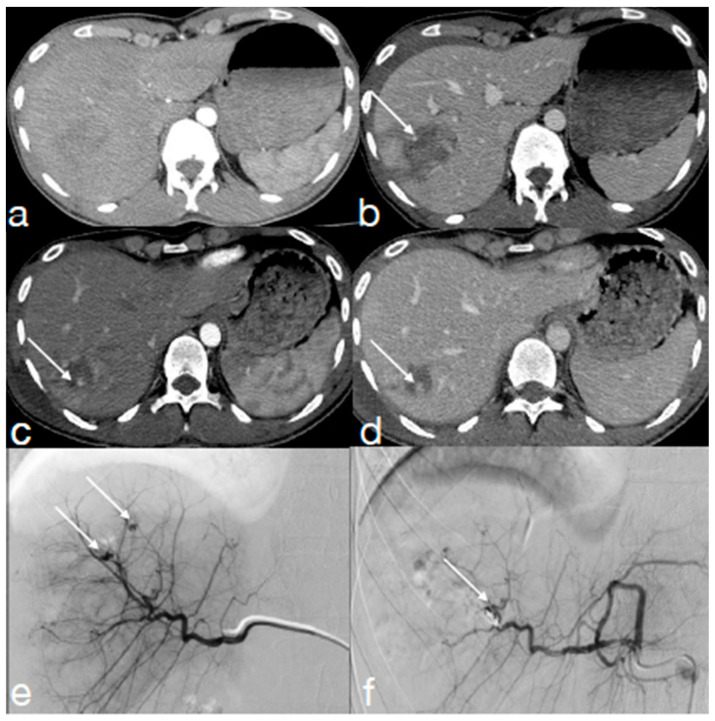
A forty-four-year-old male patient underwent high-energy blunt trauma. Enhanced CT upon admission ((**a**) arterial phase; (**b**) venous-portal phase, axial plane) shows a large liver laceration with intraparenchymal hematoma (AAST III), with a small focus of hyperdensity initially seen in the portal phase ((**b**) arrow) suspected for contained vascular injury. At the follow-up CT, which was performed 24 h later, another focus of hyperdensity suspected for pseudoaneurysm become evident ((**c**) arterial phase; (**d**), venous-portal phase, arrows). Findings were confirmed at the following angiography ((**e**) arrows) and successfully embolized ((**f**) arrow, coils).

**Figure 6 diagnostics-13-01347-f006:**
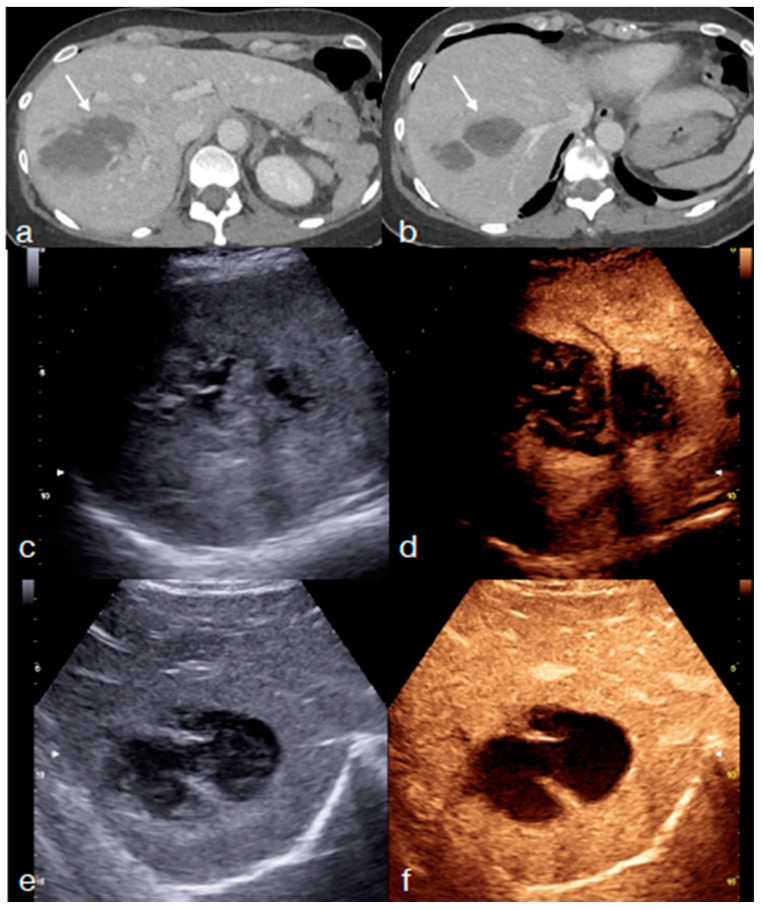
Same patient from Figure 1: once the early lesions stability was confirmed—(**a**,**b** arrows) enhanced-CT re-evaluation at 72 h from admission showing an initial healing of the liver injury—it is possible to adopt an imaging method focused on the re-evaluation of the liver injury (most relevant among the patient’ injuries; in this case, CEUS), allowing an optimal parenchymal study. CEUS was performed at 5th ((**c**) B-mode and (**d**) CEUS) and 10th ((**e**) B-mode and (**f**) CEUS) days showing the progressive healing of the liver laceration, most irregular and inhomogeneous at 5th day (**c**,**d**), becoming more contained and defined at the 10-day follow-up (**e**,**f**).

**Figure 7 diagnostics-13-01347-f007:**
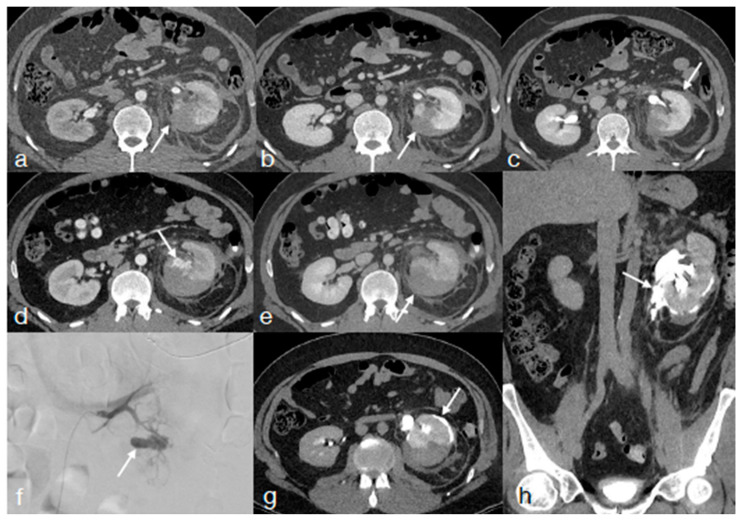
A twenty-two-year-old patient underwent high-energy blunt trauma, bike accident. Admission CT in arterial (**a**), venous (**b**), and late (**c**) phases showing a left kidney high-grade lacerations ((**a**–**c**), arrows) without any vascular or excretory injuries. At the follow-up CT performed 48 h later ((**d**) arterial phase and (**e**), venous phase), it become evident a contained vascular injury ((**d**) arrow) detectable in the arterial phase of the study ((**d**), arrow), confirmed at the subsequent angiography ((**f**), arrow) and embolized. At the following CT performed at about 1 week, a urinary leak became evident (**g**,**h** late phase, arrows) not seen at the admission CT.

**Figure 8 diagnostics-13-01347-f008:**
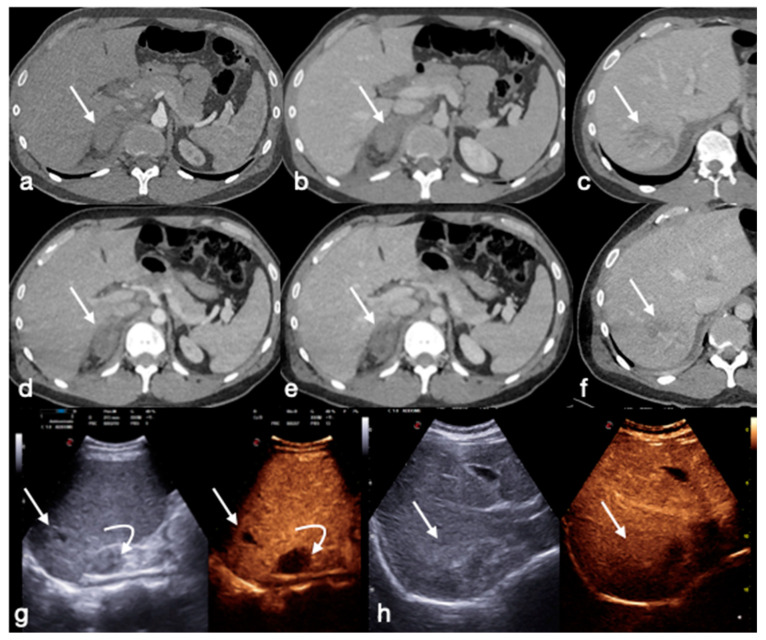
Thirty-five-year-old male patient with right adrenal hematoma and liver laceration after blunt trauma: (**a**–**c**) admission enhanced CT in arterial (**a**) and venous phases (**b**,**c**) showing the adrenal hematoma (AAST V; (**a**,**b**) arrows) and the liver laceration (AAST II, (**c**) arrow). (**d**–**f**) Follow-up CT performed five days after trauma, showing stable findings ((**d**,**e**) adrenal hematoma, arrows; (**f**) liver laceration, arrow). Subsequently, the patient underwent CEUS examination (**g**,**h**) demonstrating the progressive healing of the liver laceration ((**g**,**h**) straight arrows) and the slow resorption of adrenal hematoma ((**g**) curved arrow).

**Figure 9 diagnostics-13-01347-f009:**
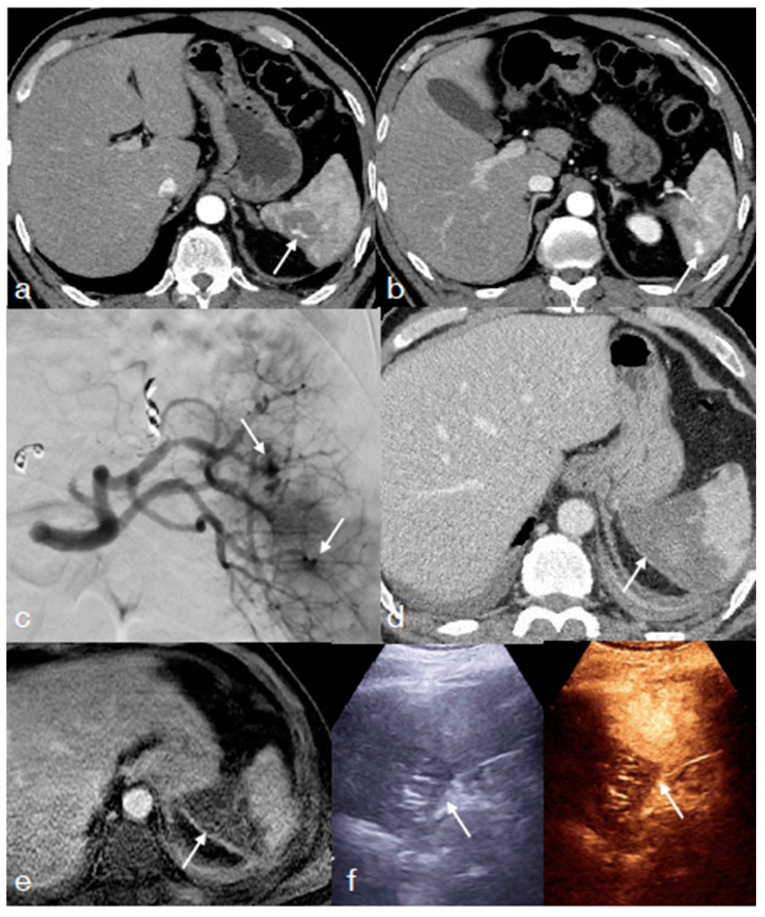
A fifty-year-old male patient underwent high-energy trauma and developing multiple contained vascular splenic injuries ((**a**,**b**) arrows) confirmed at the following angiography ((**c**) arrow) and embolized. In the subsequent follow-up, the patient was initially evaluated by enhanced CT, which detected the infarcted area due to embolization ((**d**) arrow), and in the following days, by MRI ((**e**) arrow) and CEUS ((**f**) arrows), documenting the progressive healing of the lesion.

**Figure 10 diagnostics-13-01347-f010:**
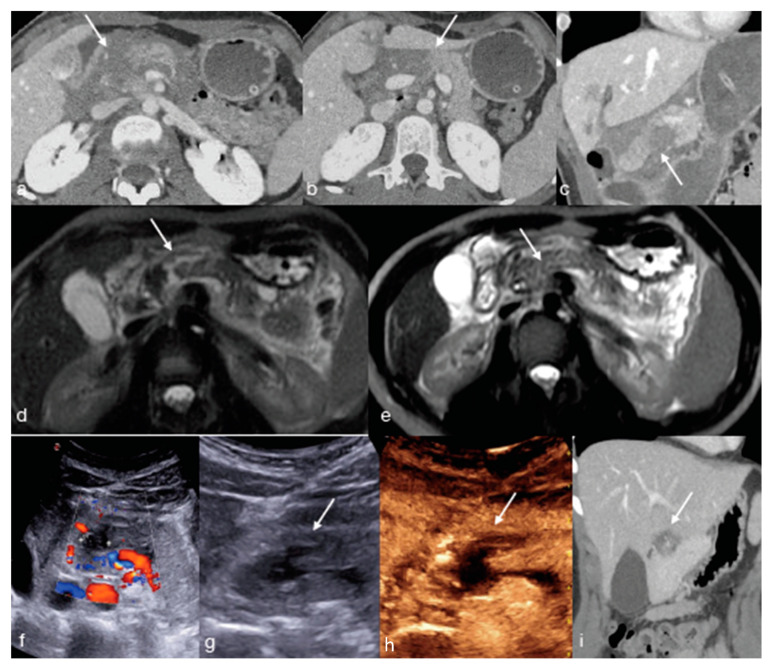
A twenty-two-year-old female who fell down the stairs: (**a**–**c**) enhanced CT in axial (**a**,**b**) and coronal oblique view (**c**) showing a pancreatic laceration (**a**–**c**, arrows). The day after, a MRI was performed to evaluate the main duct involvement ((**d**,**e**) arrows) and a lesion was excluded (AAST II), so the patient underwent a subsequent follow-up at 5 days by US-Doppler (**f**) and CEUS ((**g**,**h**) arrows), and at 20 days by CT ((**i**) coronal oblique view, arrow).

**Figure 11 diagnostics-13-01347-f011:**
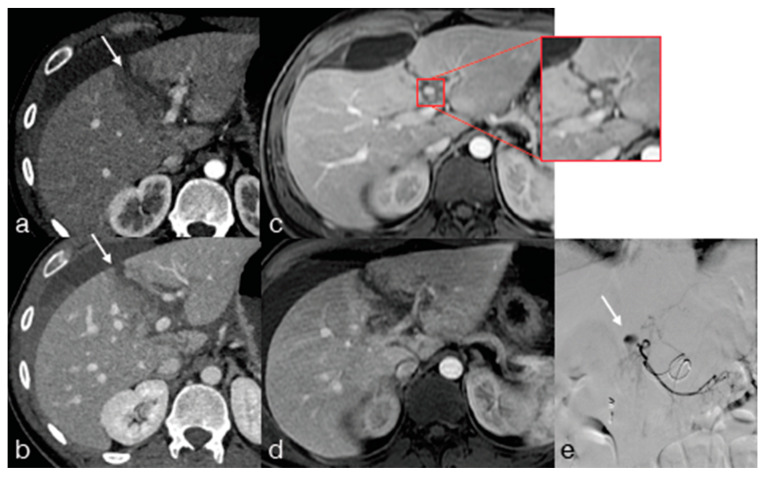
Thirty-four-year-old male underwent high-energy trauma with high-grade liver laceration. Enhanced CT upon admission for arterial (**a**) and portal (**b**) phase showing the liver injury (**a**,**b**) arrow), and at the MR follow-up performed 14 days later, the patient developed an arterial pseudoaneurysm ((**c**), T1W Fat Sat, arterial phase, red box) seen only in the arterial phase ((**c**) red box and (**d**) portal phase), which was confirmed and embolized at the following angiography ((**e**), arrow).

**Table 1 diagnostics-13-01347-t001:** Advantages and recommendations, and disadvantages and contraindications of the imaging technique that can be adopted in the follow-up of polytraumatized patients [41,45]. CECT (contrast enhanced CT); CEMRI (contrast enhanced MRI); MPR (multiplanar reconstruction); MIP (maximum intensity projection).

Follow-Up Diagnostic Technique Abdominal Parenchymal Trauma	Advantages and Recommendation	Disadvantages and Contraindications
CECT	Widespread technique Fast scan acquisition Radiation dose and contrast media reduction protocol CT scan (dual-energy CT scan-improvement CT technology) High spatial resolution MPR and MIP reconstruction utility Associated neck, thoracic, mesenteric and bowel injuries Urinary trauma/urinary leak	Radiation dose Renal insufficiency To be limited, when possible, in young age/fertile age patient/pregnant patients Iodinate contrast medium adverse reactions Not useful in biliary leak detection Radiation beam artefacts in uncooperative patients and adducted arms
CEUS	No radiation dose Young-age/fertile-age/pediatric-age (off label) patient safety No blood test before contrast-agent injection (in renal insufficiency is recommended) Low-rate adverse reaction No CT and MRI contrast media crossover (different contrast media excretion) Further evaluate uncertain CT or MRI findings related to trauma (point-of-care CEUS) Contrast-enhancement parenchymal injuries revaluation with the same dose of contrast-media administration (flash mode)	Operator expertise No MPR or MIP reconstruction No panoramic exam (reduced field of view due to meteorism or morphotype of the patient) Not useful in urinary, thoracic and bowel injuries Not useful in biliary leak detection It is preferable to avoid the injection of US contrast agent in pregnant patients; for breastfeeding women, it is recommended to abstain from breastfeeding for a period of 1–2 h Unstable ischemic disease and severe pulmonary hypertension
MRI/CEMRI	No radiation dose Young age/fertile age/pediatric age patient safety MPR acquisition/reconstruction and MIP reconstruction utility Pregnancy (preferably without contrast-medium injection) Renal insufficiency Urinary-trauma/urinary-leak detection Biliary leak detection (gadoxetate disodium contrast media is recommended/off-label use) Multiparametric exam (the use of contrast media could be not always necessary in the long-term follow-ups) Further evaluate uncertain CT findings related to cross-beam artefact	Not widespread technique and operator expertise in this field Long scan times protocol (scan time optimization is needed) Low spatial resolution Gadolinium adverse reactions No MRI conditional medical element (prosthesis, pacemaker and more) Claustrophobic or uncooperative patients Movement artefact Not useful in thoracic and bowel injuries Iv cm is relatively contraindicated during pregnancy

## Data Availability

Not applicable.

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
