# Peer review of "Non-Operative Management of Polytraumatized Patients: Body Imaging beyond CT"

_diagnostics, 2023, doi:10.3390/diagnostics13071347_

Round 1

Reviewer 1 Report

This is an interesting review on the nonoperative management of polytraumatized patients. The manuscript is focused on the role of contrast-enhanced CT, US and MRI discussing their advantages and limitations in patients with trauma. The figures are adequate and clear. However, I have some comments that should be considered for this manuscript:

-Computed tomography, page 2: “Computed Tomography (CT) represents the “gold standard” imaging technique in polytraumatized patient”. This sentence is repeated two times in the two subsequent paragraphs. I suggest to change it.

-Table 1: “Not authorized for breast feeding women (abstain from breast-feeding is recommended for a period of 12-24 hours)”. This is not true considering current guidelines. Please see ESUR guidelines (page 30 at https://www.esur.org/wp-content/uploads/2022/03/ESUR-Guidelines-10_0-Final-Version.pdf). While pregnancy cannot be included in the “advantages” of CT as it is should be performed with caution in very major trauma. Please revise advantages and limitations in the table.

-In my opinion there is an excessive number of self-citations; for instance, the first Author of this review is present in 18 references in this article, which may be too much.

Author Response

The authors would like to thank the Reviewer for its useful suggestions, helpful to improve manuscript quality.

This is an interesting review on the nonoperative management of polytraumatized patients. The manuscript is focused on the role of contrast-enhanced CT, US and MRI discussing their advantages and limitations in patients with trauma. The figures are adequate and clear. However, I have some comments that should be considered for this manuscript:

-Computed tomography, page 2: “Computed Tomography (CT) represents the “gold standard” imaging technique in polytraumatized patient”. This sentence is repeated two times in the two subsequent paragraphs. I suggest to change it.

Ok, the second sentence was changed as follow: “Computed Tomography (CT) constitutes the reference imaging method in polytraumatized patient, mainly due to the very high diagnostic performance for whole body injuries in short time, when an up-to-date technology is used and a correct imaging protocol is adopted, despite suffering from invasiveness due to the radiation dose and to intravenous contrast agent administration

-Table 1: “Not authorized for breast feeding women (abstain from breast-feeding is recommended for a period of 12-24 hours)”. This is not true considering current guidelines. Please see ESUR guidelines (page 30 at https://www.esur.org/wp-content/uploads/2022/03/ESUR-Guidelines-10_0-Final-Version.pdf). While pregnancy cannot be included in the “advantages” of CT as it is should be performed with caution in very major trauma. Please revise advantages and limitations in the table.

I agree, advantages and limitation of CT use were revised in the table.

-In my opinion there is an excessive number of self-citations; for instance, the first Author of this review is present in 18 references in this article, which may be too much.

It’s true that there are multiple citation of our research group and this is related with the topics addressed in the article which, as can be verified from the cited  references, has been the subject of research of our group for a long time.

Reviewer 2 Report

This is a case-based review of the imaging techniques following trauma.

I have few important remarks that should be imbedded into the review:

1. The type and need of follow-up studies in multitrauma patients are dictated by clinical suspicion and hemodynamical status of patients. Most low-grade injuries (any of abdominal or others) does not require any follow-up radiological study.

2. F-U imaging should be based on local expertise and availability.

3. The use of f-u studies is not a “must” as mentioned, but an additional factor in clinical decision- making. The example – Figure 4, when early rescanning was probably unnecessary.

I would suggest changing the Title: case-based review, follow-up options, abdominal trauma (maybe also chest).

Author Response

The authors would like to thanks the Reviewer for its useful suggestions, helpful to improve the manuscript quality.

This is a case-based review of the imaging techniques following trauma.

I have few important remarks that should be imbedded into the review:

  1. The type and need of follow-up studies in multitrauma patients are dictated by clinical suspicion and hemodynamical status of patients. Most low-grade injuries (any of abdominal or others) does not require any follow-up radiological study.

We agree and this was reported in the article [added in the introduction: “Computed Tomography (CT) represents the “gold standard” imaging technique in the first line evaluation of polytraumatized patients [1-3] as it ensure high diagnostic accuracy, rapid execution time and hospital availability, however, once the diagnosis is made and the patient is stable, a series of possibilities open up regarding the best diagnostic imaging tool to choose for monitoring the diagnosed lesions or detecting complications, depending on the involved anatomical structures, the grade of each injury and the availability and expertise of each imaging method [4-9]], as well as the indications for the imaging follow-up of injuries, according with available guidelines”].

2. F-U imaging should be based on local expertise and availability.

We agree and this was added in the introduction: “Computed Tomography (CT) represents the “gold standard” imaging technique in the first line evaluation of polytraumatized patients [1-3] as it ensure high diagnostic accuracy, rapid execution time and hospital availability, however, once the diagnosis is made and the patient is stable, a series of possibilities open up regarding the best diagnostic imaging tool to choose for monitoring the diagnosed lesions or detecting complications, depending on the involved anatomical structures, the grade of each injury and the availability and expertise of each imaging method [4-9]], as well as the indications for the imaging follow-up of injuries, according with available guidelines”

3. The use of f-u studies is not a “must” as mentioned, but an additional factor in clinical decision- making. The example – Figure 4, when early rescanning was probably unnecessary. 

I agree that the use of f-u studies is not a “must”, the indications reported for the timing of the imaging follow-up of polytraumatized patients are in accordance with available guidelines and clinical experience.

In the reported example of Figure 4 the CT re-evaluation was required to understand the reason of the new onset of abdominal symptoms, with may be related with different possibilities…

I would suggest changing the Title: case-based review, follow-up options, abdominal trauma (maybe also chest)

Ok, thanks for the suggestion, the title was modified as follow: Non-operative management of polytraumatized patients: body imaging beyond CT. How, why and when. A case-based review.

Round 2

Reviewer 1 Report

The revised manuscript is overall satisfactory. 

Author Response

Thank you.

Best regards,

The Authors

Reviewer 2 Report

Small revisions:

1. Fig 2 – When F-U CT was done (24, 48 hours)?

2. Fig 2 and 3 are very similar. Explain the difference and if no – delete one of them.

3. Add limitation for conclusion.  

To our knowledge follow-up imaging modality in the NOM setting is still debated. The results of many papers showed that a routine imaging follow-up, ultrasound and CT scan, has limited therapeutic advantage in terms of predicting failure of NOM. Indication for follow-up imaging should be based on clinical findings. If indicated, a CT scan should be used as preferred imaging modality.

Author Response

1-2) Yes, the cases are similar. We delete Figure 2.

3) We agree, as stated in the Figure 3 legend, and we added Your suggestions in the conclusion, changed as follow: "

Follow-up imaging timing and modality in the NOM setting is still debated. However, in adult polytraumatized patients after blunt trauma with body injuries, particularly with abdominal parenchymal injuries non-operatively managed, the adoption of multiple, integrated imaging methods may offer good opportunities in the follow-up, when clinically indicated, adding the potentialities of each one and reducing the limits and negative effects."